# Spatial-Temporal Changes and Simulation of Land Use in Metropolitan Areas: A Case of the Zhengzhou Metropolitan Area, China

**DOI:** 10.3390/ijerph192114089

**Published:** 2022-10-28

**Authors:** Xiuyan Zhao, Changhong Miao

**Affiliations:** 1Key Research Institute of Yellow River Civilization and Sustainable Development & Collaborative Innovation Center on Yellow River Civilization Jointly Built by Henan Province and Ministry of Education, Henan University, Kaifeng 475001, China; 2College of Geography and Environmental Science, Henan University, Kaifeng 475004, China

**Keywords:** land use change, CA–Markov, MCE model, Zhengzhou metropolitan area, driving forces

## Abstract

Metropolitan areas are the main spatial units sustaining development. Investigating internal factor changes in metropolitan areas are of great significance for improving the quality of development in these areas. As an emerging national central city of China, Zhengzhou has experienced rapid urban expansion and urbanization. In this study, principal component analysis and the model and Geodetector model were used to comprehensively analyze the influencing factors of land use change in Zhengzhou from 1980 to 2015. Based on the CA–Markov model, we improved the accuracy of multi-criteria evaluation of suitability factors and simulated land use change in 2015. The results show that land use conversions in the study area between 1980 and 2015 were frequent, with the areas of farmland, woodland, grassland, water, and unused land decreasing by 5.00%, 17.12%, 21.59%, 18.31%, and 94.48%, respectively, while construction land increased by 53.61%. The key influences on land use change are the urbanization and growth of residential or non-agricultural populations. In 2035, the area of farmland in the study area will decrease by 11.09% compared with that in 2015 and construction land will increase by 38.94%, while the area of other land use types will not significantly change. Zhengzhou, as the center city, forms a diamond-shaped core development area of Zhengzhou–Kaifeng–Xinxiang–Jiaozuo, while Xuchang is considered an independent sub-center uniting the surrounding cities for expansion. With its radiation power of unipolar core development for many years and the developmental momentum of Zhengzhou–Kaifeng integration, Zhengzhou city jointly drives the economic development of the surrounding cities. The protection of farmland and control of the expansion of construction land are the major challenges for the Zhengzhou metropolitan area to achieve sustainable development.

## 1. Introduction

Land use change is one of the main research elements of global environmental change and sustainable development [1,2,3,4]. It is not only a visual representation of land use pattern change but also affects ecological, environmental, and regional climatic changes [5,6,7,8]. Therefore, modeling and predicting land use changes and exploring their internal spatial and temporal mechanisms and rules are important for guiding future regional development and controlling the direction of land use development to achieve the goal of sustainable development.

Land use change has been a hot topic, and scholars have conducted different studies in this field in recent years [9,10,11]. There is extensive and in-depth research on the spatial and temporal patterns [12,13,14,15], driving forces [10,16,17,18,19,20,21], and dynamic models [22,23,24] of land use change. With the development of human society and the economy, the conflict between human and land has become increasingly serious, and some scholars have paid more attention to the subject of the relationship between human and land [25,26,27]. Some research outlines the concerns of how land use changes generated by the rapid increase in the population will negatively impact on the climate [28,29,30,31] and ecological environment [32,33,34,35]. Scholars have investigated the drivers of land use change at different spatial scales in China [10,12,36,37]; among them, most have studied the relationship between farmland and construction land [17,38,39]. Overall, previous research indicates that significant land use changes have occurred globally over the past decades. Each country, region, and city have different patterns of land use and influencing factors. However, population increase and human economic development are generally considered the two most important influencing factors driving land use change [4,10,40].

In terms of methodology, the simulation and prediction of land use change are currently dominated by model simulations. Among these, artificial neural networks (ANN) [41,42,43], agent-based model (ABM) [44,45], meta-cellular automata (CA) [43,46,47], economic models, Markov chains, and machine learning models [48] are the main methods used for the simulation and prediction of land use change. Since each model has different advantages and disadvantages, there are more studies with multiple model coupling [49,50,51,52]. In recent years, more and more scholars have used urban CA to simulate real cities, mainly focusing on the simulation of urban expansion, and some scholars have tried to use CA for urban planning For example, Li et al. [53,54,55,56] proposed a CA–Markov model based on an artificial neural network and ant colony algorithm. Kheder et al. [57] combined genetic algorithms with meta-cellular automata and showed the significant effectiveness of the method. Overall, the coupled CA–Markov model effectively combines the advantages of the Markov long-term series simulation and powerful complex spatial prediction capability of CA [58,59,60], while adding a multi-criteria evaluation mechanism (MCE) to generate a land suitability atlas [61] to make the simulation and prediction results more informative.

As an important city in the central region of China, Zhengzhou city has experienced rapid economic development and drastic changes in land use in recent years [62]. With the Zhengzhou Airport economic experimental zone becoming a national strategy supported by the construction of the Zhongyuan Economic Zone, Zhengzhou, as the core city of the Zhongyuan Urban Agglomeration, has gained significant development opportunities. Against the background of rapid urban development, the land use types in Zhengzhou are frequently transformed among each other, and the contradiction between human and land gradually emerges and these changes are a dynamic process [63]. Therefore, it is of fundamental importance to study the characteristics of spatial and temporal changes in land use in the Zhengzhou metropolitan area and explore its change patterns and trends to solve current problems related to the sustainable development of megacities. In the present study, we examined the characteristics of land use change in Zhengzhou from 1980 to 2015 using remote sensing images and other auxiliary data to analyze its driving forces. Moreover, the spatial and temporal land use changes and evolution patterns in 2015 were simulated using the CA–Markov model with MCE suitability factors for predicting future changes. The results of this study can provide scientific support and a theoretical basis for land use planning and management, urban construction, and sustainable development of government departments, and they have implications for the development planning of mega-city clusters in China.

## 2. Materials and Methods

### 2.1. Study Area

The Zhengzhou metropolitan area includes five prefecture-level cities in the Henan province: Zhengzhou, Kaifeng, Xinxiang, Jiaozuo, and Xuchang (Figure 1). At the end of 2016, the Central Plains City Cluster Development Plan, approved by the State Council, proposed to support Zhengzhou to build a national central city, promote the deep integration of Zhengzhou with Kaifeng, Xinxiang, Jiaozuo, and Xuchang, build a modern metropolitan area, and form a core that drives the periphery, radiates the whole country, and connects international regions. Instead of simply cutting the Central Plains city cluster smaller, the Zhengzhou metropolitan area is more pragmatic for promoting the flow of regional factors and strengthening the economic potential and “polarization effect” of the “Zheng leader”. The abovementioned plan points out that in the future, as the radiation-driving capacity and level of integration of Zhengzhou Metropolitan continue to improve, on the basis of the current spatial scope, it will gradually include Kaifeng, Xinxiang, Jiaozuo, and Xuchang counties under the jurisdiction of the city and Ruzhou, Lankao, and other counties (cities) under the direct control of the province into the scope of the Zhengzhou Metropolitan area to accelerate the formation of a network, cluster, intensive spatial development pattern, driving the Central Plains city cluster to a national city cluster with international influence [64].

### 2.2. Data Sources

This study used the following data: (1) Land use data from 1980, 1995, 2000, 2005, and 2015 of the Henan province obtained from the Resource and Environmental Science and Data Center, Chinese Academy of Sciences (https://www.resdc.cn/, accessed on 1 July 2021); (2) ASTER GDEM V3 digital elevation at 30 × 30 m resolution (2015), population (2015), precipitation (2015), temperature (2015), and Henan provincial political subdivision data from the Geographic State Monitoring Cloud Platform in 2015 (https://www.resdc.cn/, accessed on 5 July 2021); (3) administrative divisions, roads, rivers, and other auxiliary data from the Geographic State Monitoring Cloud Platform (https://www.resdc.cn/, accessed on 18 July 2021); (4) total population and GDP data of the study area from the statistical yearbooks from each time period (http://www.henan.gov.cn/, accessed on 19 July 2021).

Land use data are published every five years, with 1980 being the first year of publication. In the CA–Markov simulation, to keep the land use transfer matrix consistent with the simulated time span, the above years are selected.

### 2.3. Data Processing

The data from each year were reunified into the Albers projection using the raster projection tool in the GIS software, and the spatial resolution was unified to 30 m. According to the Classification of Land Use Status released by the Ministry of Land and Resources, a combination of supervised and unsupervised classification was used to decode the images, and the land use types were then classified into farmland (FL), woodland (WL), grassland (GL), water area (WA), construction land (CL), and unused land (UN). The spatial data of the Zhengzhou metropolitan area from the land use data of the Henan province were used as a mask in ArcGIS. The slope function in ArcGIS was then used to process the DEM data to generate slope data and unify it into the Albers projection with a resolution of 30 m (Figure 2). The raster to ASCII function in ArcGIS was used to convert all raster data into ASCII format in IDRISI. All translated ASCII data were converted into uniform raster data in the IDRISI software. Vector data (roads and rivers) were imported into IDRISI, using land use data as the coordinate system template, and then converted into raster data. The influence range of each road was then calculated using the DISTANCE tool, and the data in the statistical yearbook were imported into Microsoft Excel to unify the format. Lastly, SPSS software was used to complete the extraction parameters.

### 2.4. Research Methods

#### 2.4.1. Land Use Change

The land use transfer matrix is a common method for measuring changes in land use types over time. Based on the land use data, the land use transfer matrix was calculated for the Zhengzhou metropolitan area in four periods, i.e., from 1980 to 1995, 1995 to 2000, 2000 to 2005, and 2005 to 2015. The equation of the land use matrix is expressed as follows:(1)$$D_{i}=\left\{\sum_{j}^{n}\left(\frac{\left|\Delta S_{i\to j}+\Delta S_{j\to i}\right|}{S_{a}}\right)\right\}\times \frac{1}{t}\times 100\%$$
where $D_{i}$ is the dynamic degree of the land use type i; $\left|\Delta S_{i\to j}+\Delta S_{j\to i}\right|$ represents the absolute value of the area converted from the i-type of land to the j-type of land plus the area converted from the j-type of land to the i-type of land; $S_{a}$ is the area of spatial unit a; t represents the length of the study period.

#### 2.4.2. Driving Forces of Land Use Change

##### GeoDetector Model

GeoDetector is a statistical method used for detecting the consistency of spatial distribution patterns between dependent and independent variables based on the geospatial divergence theory. The GeoDetector model contains four detection modules: factor, interaction, risk, and ecological detection. The factor detection module is used to examine the effect of the independent variable on the dependent variable as follows:(2)$$\begin{array}{c} q=1-\frac{\sum_{h=1}^{1}N_{h}\sigma_{h}^{2}}{N\sigma^{2}}=1-\frac{SSW}{SST}\\ SSW=\overset{L}{\underset{h=1}{\sum }}N_{h}\sigma_{h}^{2},SST=N\sigma^{2} \end{array}$$
where *h* = 1, ..., *n*; *L* is the strata of variable *Y* or factor *X*, which is the classification or partitioning; $N_{h}$ and *N* are the h and full area of the number of cells; $\sigma_{h}^{2}$ and $\sigma^{2}$ are the variance of the *h* and *Y*-values, respectively; SSW and SST are the “within sum of squares” and “total sum of squares”; the value range of *q* is [0, 1].

In this study, the GeoDetector was used to detect the impacts of relevant factors and their interactions in influencing land use change in the Zhengzhou metropolitan area from 1980 to 2015.

##### Patch-Generating Land Use Simulation (PLUS) Model

The PLUS model was constructed based on the rule mining framework of the Land Extension Analysis Strategy (LEAS) and CA model of multi-type random patch seeding (CARS). This study used LEAS to analyze the factors that influenced the changes in land use types between 1980 and 2015.

#### 2.4.3. Simulation

##### Cellular Automata (CA)

Unlike general dynamical models, meta-cellular automata are not determined by strictly defined physical equations or functions but are composed of a set of rules for model construction. CA is a grid dynamical model, with spatio-temporal computational capabilities with discrete time, space, state, and local influence of all cells, used to simulate the spatio-temporal evolution of complex systems in GIS [65]. Therefore, “meta-cellular automata” is a general term for a class of models or methodological framework. It is characterized by discrete time, space, and state (taking only finitely many states for each variable), and its state-change rules are local in time and space. CA systems consist of a meta-cell, state, neighborhood, and transformation rule, usually applying a 5 × 5 Moore neighborhood as a filter following the transformation rule, i.e., that the effect on the state transformation of the central meta-cell is inversely proportional to the distance from the central meta-cell. The traditional CA transition rules only consider the influence of the neighborhood, and each tuple is converted based on local rules. This model is represented by Wang et al. [66].
(3)$$S_{t+1}=f(S_{t},N),$$
where $S_{t+1}$ is the finite, discrete set of states of the meta-cell; f is the meta-cell transformation rule for the local space; *t*, *t* + 1 denote different moments; *N* is the neighborhood of the meta-cell.

The factors affecting land use change to construction land are often nonlinear, and it is difficult to accurately simulate real urban development by relying solely on the local conversion rules of traditional CA [51,66,67]. Therefore, in this study, the conversion rules of CA were constrained and controlled by three elements: the local influence of the meta-cell neighborhood, global constraint of the Markov chain control, and suitability of the meta-cell.

##### Markov Model

A Markov chain is a discrete-time stochastic process with Markovian property in mathematics. The Markov process is a process in which a finite time series, $t_{1}<t_{2}<t_{3}<\cdots t_{n}$, within a finite time series, is at any moment, $t_{m}$ of the state $a_{m}$, only related to the $t_{m-1}$ state at the moment of time $a_{m-1}$ and independent of the $t_{m-1}$ state at the previous moment. The process is such that the past is irrelevant for predicting the future given the current information. The Markov processes in which both time and state are discrete are called Markov chains [68]. Transitions between areas of land use types can be viewed as probabilities of state shifts, and the probability matrix can be calculated using the following equation:(4)$$P_{ij}=\left[\begin{array}{cccc} P_{11} & P_{12} & \cdots & P_{1n}\\ P_{21} & P_{22} & \cdots & P_{2n}\\ \vdots & \vdots & \vdots & \vdots \\ P_{n1} & P_{n2} & \cdots & P_{nn} \end{array}\right]$$
$$0\leq P_{ij}<1\;\mathrm{and}\sum_{j-1}^{n}P_{ij}=1\left(i,j=1,2,\cdots ,n;\right)$$

The changes in land use structure can be further calculated by means of the land use transfer probability matrix:(5)$$S\left(T\right)=P_{ij}\times S\left(T_{0}\right)$$
where $S\left(T\right)$ and $S\left(T_{0}\right)$ are the T and $T_{0}$ state of the land use structure at the moment, respectively, and $P_{ij}$ is the land use type transfer probability matrix.

##### Multi-Criteria Evaluation (MCE) Method

Considering the role of multiple factors in land use type conversions, MCE is used to quantify non-linear factors, such as natural and socio-economic factors, that are difficult to quantify [69]. MCE analysis involves two types of criteria, i.e., constraints and constraint factors, which are the Boolean criteria that limit the analysis to a specific area and define the degree of suitability for each image element. In this study, the hierarchical analysis method (AHP) was applied to pair the constraints of different land categories, compare the weights of each factor, and integrate each standardized factor with its corresponding weight and superimpose it with the constraints by the weighted linear combination method (WLC) to obtain the suitability distribution of each land category as a supplement to the local rules of CA. The addition of suitability maps to the CA conversion rules, as the suitability elements of the meta-cell itself, changes the situation in which traditional CA relies on local rules for simulation and is consistent with the complex nonlinear characteristics of urban land use type changes.

According to the geographic conditions of the Zhengzhou metropolitan area and the characteristics of land resource utilization, construction land and water area were selected as the constraints, where slope and distance from water area and road were used as the constraints. The results of each suitability evaluation were based on the data of the starting year of the simulation, and the distance factors, such as slope, water, and road, were all static data. In addition, since the constraint factors were the same for each category, the weight assignments were the same for different years of the simulation. Since the selected constraint factors had different units and scales, they were stretched to continuous values from 0 to 255 (with gradually increasing suitability) using the fuzzy affiliation function and control points and then linearly integrated with their corresponding weights [70]. Lastly, the suitability distribution maps of individual land classes were obtained.

##### CA–Markov Model and Prediction Method

The CA–Markov model combines the advantages of the Markov model in time dimension analysis and CA model in spatial dimension analysis, which reduces the difficulty of making conversion rules and interference of human factors. Based on the CA–Markov model in the IDRISI software, the prediction simulation of land use pattern changes in the study area was carried out as follows:

(1) Calculation of land use transfer probability matrices: The land use transfer probability and area matrices were calculated from the Markov chain model using four sets of data for the overlay analysis, i.e., 1980, 2015; 1995, 2015; 2000, 2015; and 2005, 2015, where the probability matrices were used as transformation rules to provide support for ensuing the CA–Markov simulation operations.

(2) Create suitability atlases: The MCE module in the IDRISI software was used to create individual suitability maps for each of the six land types in each phase of the image, which were then combined into a suitability atlas. Consider the suitability map of farmland as an example. Based on the realistic land use type conversion pattern, the parameters of slope, elevation, distance from major highways and railroads, and the data of watershed, ecological and arable red line, and basic farmland were selected to debug the constraints. According to the grading standard for farmland, the S-type decreasing function of the fuzzy affiliation function was selected to stretch the slope interval to [0°, 15°] and elevation interval to 0–300 m, and the step-type monotonic decreasing function was selected to set the distance above 100 m from the highway and above 100 m from the railroad; the area within the red line of farmland does not change. The AHP method in MCE was used to obtain the weight assignment of each constraint factor in the simulation [70].

(3) Parameter setting: Consider 1980 and 2015 as an example. Using 2015 as the base year, a 5 × 5 meta-cell filter was set, and the number of meta-cell simulation cycles was adjusted to 35.

(4) Land use change simulation accuracy testing: The Kappa coefficient can test the degree of consistency between simulation results and real data and is widely used in the study of simulation accuracy testing. The Kappa coefficient is calculated using the following formula:(6)$$Kappa=\frac{P_{o}P_{c}}{P_{p}P_{c}}$$
where $P_{o}$ is the proportion of correctly simulated raster cells, $P_{c}$ is the proportion of correctly simulated raster cells in the random case, and $P_{p}$ is the proportion of correctly simulated raster cells in the ideal condition. The test results of the Kappa coefficient are in the interval of [−1, 1], and the classification evaluation criteria of the Kappa coefficient are as follows: (−∞, 0.00) is poor; (0.00, 0.20) is weak; (0.20, 0.40) is weak; (0.40, 0.60) is moderate; (0.60, 0.80) is significant; and (0.80, 1.00) is the best.

The analysis framework can be seen in Figure 3.

## 3. Results

### 3.1. Analysis of Land Use Change

#### 3.1.1. Land Use Change

The land use changes in the Zhengzhou metropolitan area were analyzed from 1980 to 2015 (Figure 4). The results show that the main changes in land use types in the Zhengzhou metropolitan area were the interconversion of farmland and construction land, woodland and farmland, and watershed and farmland. Land use changes in the Zhengzhou metropolitan area had distinct characteristics in two separate periods. Overall, the farmland area in the area decreased by 1100 km² between 1980 and 2005; this was mainly due to the high rate of urbanization and encroachment of a large amount of farmland by construction land. The area of woodland and grassland decreased by 307 km² and 365 km², respectively, and most of these areas were reclaimed as farmland. Between 2005 and 2015, farmland and construction land were converted into each other, but the area of each land type remained almost unchanged.

The area of each land use type in the Zhengzhou metropolitan area from 1980 to 2015 and overall trend changes are shown in Figure 5. The land use types in the study area, in descending order of area, are farmland, construction land, woodland, grassland, water area, and unused land. Among these, farmlands increased slightly from 24,923.91 km² to 25,815.55 km² between 1980 and 1995, declined to 25,115.62 km² by 2000, and continued to decline to 23,679.37 km² in 2015. Woodlands increased slightly from 2041.37 km² to 2364.79 km² between 1980 and 1995, declined to 2001.99 km² in 2000, and decreased markedly to 1691.96 km² in 2015. Grasslands decreased considerably from 1900.21 km² to 1459.23 km² between 1980 and 1995, increased markedly to 1863.93 km² by 2000, and then decreased considerably to 1490.03 km² in 2015. Watershed areas decreased drastically from 1180.32 km² to 697.61 km² between 1980 and 1995 and increased slightly to 713.10 km² by 2000 and then again to 964.22 km² by 2015. Construction land decreased slightly from 4290.60 km² to 4078.87 km² between 1980 and 1995, increased to 4674.78 km² by 2000, and markedly increased to 6590.93 km² in the period 2000–2015. The unused land decreased tremendously to 7.43 km² during the period 1980–1995, increased to 54.06 km² in 2000, and then decreased to 6.98 km² in 2015.

The area of farmland in the study area decreased considerably over 35 years, from 24,923.91 km² to 23,679.37 km², with the proportion decreasing from 72.32% to 68.79%, indicating that in the context of economic development, some farmland resources were occupied by other lands in the process of urbanization. The area of woodland showed a growing trend before 2000, which was then followed by a significant decline from 2041.37 km² to 1691.96 km², with a total percentage decrease of 1%. The grassland area showed a similar situation to woodlands, decreasing from 1900.21 km² to 1490.03 km², with a land use change total share of 1.18%. The area of the water area land cover type decreased drastically from 1180.32 km² to 964.22 km², with a percentage decrease from 3.42% to 2.80%. The fastest growth occurred in the area of land for construction, from 4290.60 km² to 6590.93 km², with an increase of 7.05%. The area of unutilized land decreased by 119.34 km² and the proportion decreased by 0.02%.

The dynamic attitude of land use in the study area from 1980–2015 is shown in Table 1 and Figure 6. The overall activity of land use change within the study area was high, with the most significant activity observed for unused land, which reached 44.96% between 2000 and 2015. However, an even larger decrease of 106.74% was noted between 1980 and 1995, which indicates that unused land has been developed, utilized over time, and shifted to other land use types. Construction land showed the most significant increasing trend from 0.34% to 1.93%, followed by the woodland areas from 0.91% to 1.22%, while farmlands showed a weaker increasing trend. Grasslands, water areas, and unused land showed varying degrees of decreasing trends. The overall activity of land type transfers of woodlands, grasslands, water areas, and unused land decreased after 2005, and the area of construction land decreased considerably, while the area of farmlands increased by the same magnitude, which indicates that the areas of farmlands were protected during urban expansion due to the changes in government planning and balance between the ecological environment and urban development in recent years.

#### 3.1.2. Land Use Transfer Characteristics

The change of land use area in Zhengzhou metropolitan area from 1980 to 2015 is shown in Figure 7, and the land use transfer matrix is shown in Figure 8 and Table 2.

During the period of 1980–2015, the main transfer of land use was into construction land, in which the area of farmlands transferred to construction land was the largest, reaching 2644.23 km² and accounting for 86.39% of the total transfer of farmlands and 89.57% of the total transfer to construction land. The area of construction land that was transferred to farmlands ranked second, reaching 829.22 km² and accounting for 94.30% of the total area transferred from construction land and 42.29% of the total area transferred to farmlands. The main flow direction of woodlands was into farmlands and construction land, reaching 351.69 km² and 123.26 km², respectively, and accounting for 69.28% and 24.28%, respectively, of the total transferred area. The main flow direction of grasslands was also into farmlands and construction land, reaching 282.77 km² and 128.94 km², respectively, and accounting for 59.79% and 27.26%, respectively, of the total transferred area. The main conversion of unused land was into farmlands, accounting for 74.95% of the total converted area.

Over the past 35 years, the farmland, woodland, grassland, water area, and unused land all showed a negative net transfer of reaching 1099.91 km², 307.16 km², 366.13 km², 192.28 km², and 107.38 km², respectively. The construction land has increased significantly by 2072.86 km². It indicates that the urbanization process of construction land occupies a large number of other types of land.

### 3.2. Analysis of Land Use Change Factors

#### 3.2.1. Principal Component Analysis

In this study, principal component analysis was used to reduce the dimensionality of each complex factor that is significantly correlated within the study area, from which a few mutually independent comprehensive factors are selected that not only express their original information, but also reflect the comprehensive information within the study area. Based on the actual situation within the study area, this study selected 10 of the socio-economic factors to represent the elements of the driving force analysis, which included the following: the total population of the Zhengzhou metropolitan area, gross product of the study area, gross value of primary industries, gross value of secondary industries, gross value of tertiary industries, social fixed assets, the share of the tertiary sector in the gross regional product, total retail sales of social consumer goods, disposable income per capita of rural residents, and per capita disposable income of urban residents.

In the principal component analysis, the eigenvalues, contribution rates, and cumulative contribution rates of the first two principal components were calculated using the SPSS software. From the results (Table 3), the eigenvalue of the first principal component was 8.1911 with a variance contribution of 81.91%, and the eigenvalue of the second principal component was 1.6537 with a variance contribution of 16.54%, resulting in a cumulative variance of 98.45%. This indicates that the first two principal components can represent the initial 10 factors for further analysis. To analyze the practical significance of the principal components, the driver principal component loading matrix was used (Table 4). The correlations between each driver and principal component were strong, and the ranking, in order of strength, was as follows: social fixed assets, gross product, output value of tertiary industry, output value of secondary industry, total retail sales of consumer goods, per capita disposable income of urban residents, per capita disposable income of rural residents, total population, share of tertiary industry in regional GDP, and output value of primary industry. These factors reflect the driving forces of economic and population growth and urbanization on land use changes in the study area.

Table 4 shows that the greatest influence on land use change was the increase in the urbanization rate, resident population, or non-agricultural population. The influence of the non-agricultural population on the evolution of construction land was evident, indicating that with the increase in the urbanization level, fixed asset investment, and non-agricultural population, the demand for construction land also gradually increased, as noted by the increasing trend. The urbanization rate markedly influenced the conversion of farmlands into other land use types; this indicates that economic development and the increase in the urbanization level accelerate the shrinkage of farmland areas, which results in severe challenges. In addition, the concentration of permanent populations in cities results in the continuous expansion of settlement areas, which contributes to the reduction of farmland areas. The factors influencing the characteristics of construction land and farmland changes show that the most influential factors are the urbanization rate or growth of resident or non-agricultural populations.

The gross product, social fixed assets, total retail sales of consumer goods, per capita disposable income of urban residents, and per capita disposable income of rural residents in the Zhengzhou metropolitan area rose from 1980 to 2015, and the gross product increased from 7959 billion yuan to 1498.963 billion yuan. With the development of the economy, the land use structure of the study area has changed accordingly, with a gradual decrease in farmlands and substantial increase in construction land, which is a true reflection of the increasing level of urbanization. The increasing population has gradually reduced the land area occupied per capita, thus forcing part of the agricultural population to shift from agricultural production to other non-agricultural production land uses, which, likewise, accelerates the development of urbanization. From the analysis of the results, it can be concluded that a large part of urban and rural construction land is transformed from farmland, and the urbanization construction and industrial structure adjustment and optimization will inevitably increase the pressure on the flow of farmland, resulting in a further contradiction between urban land and farmland. With the increase in the national GDP and per capita disposable income, the highly developed big cities seem to have become the necessary carriers in the new values of people. Although policy support has led to the protection of the red line of farmlands and ecological areas, the balance between the ecological environment and economic development remains an issue that needs to be discussed in the foreseeable future.

#### 3.2.2. Impact Factor Analysis

The PLUS model was used to analyze the driving forces of land use change in the Zhengzhou metropolitan area. The results (Figure 9) show that the main drivers of farmland expansion are population growth, GDP, and distance from primary roads. Superimposed data of the increased farmland area and population raster (Figure 10e) show that the increased farmland area was mainly distributed in the beach areas on the north and south banks of the Yellow River as well as the areas with relatively small population densities, whereas the more densely populated areas had a lower increase in farmland. In order to identify the reasons for the conversion of farmlands, it is also necessary to analyze the factors influencing the reduction of these areas. In the last 35 years, approximately 14.54% of the total transfer of farmlands has been to construction land and woodlands. Among these, the area of farmlands that has been transferred to construction land accounts for 90.11% of the increase in construction land, and the amount transferred to woodlands accounts for 63.26% of the increase in woodlands. This indicates that the transformed of farmland area is the main reason for the increase in the area of construction land and woodlands. Therefore, the analysis of the drivers of the expansion of construction land and woodland area can approximate the reasons for the decrease in regional farmland area. From Figure 9 and Figure 10, it can be seen that the GDP had the strongest influence on the expansion of construction land area, followed by the population and distance from primary roads. The areas of increasing construction land were primarily located in the peripheral areas of Zhengzhou, which are relatively flat, densely populated, and have high economic development. The main driving factors affecting the increase in the woodland areas were the GDP, followed by slope, precipitation, and DEM. Most of the areas with increased woodlands were mainly concentrated in areas located a great distance away from the city. Therefore, the decrease in the farmland areas may have been influenced by factors, such as population, elevation, and GDP. Essentially, the factors driving the conversion of farmlands in Zhengzhou Metropolitan from 1980 to 2015 can be attributed to the influence of socioeconomic factors, such as population, GDP, and roads, while natural environmental factors such as elevation and precipitation are also the main influencing factors.

Since 1980, farmlands in the Zhengzhou metropolitan area have shown a decreasing trend. Most of the farmlands have been transferred into construction land and woodlands. Figure 10e shows the relationship between the shift from farmlands to woodlands and farmlands to construction land and the elevation. It can be observed that the driving factors not only influenced the spatial change of farmlands, but also limited the form of spatial change of farmlands. In the relatively flat peri-urban areas, farmlands were converted into building land on a large scale, with concentrated spatial distribution. In mountainous areas, farmlands were mainly transferred into woodlands with fewer large patches; the transferred areas were mostly scattered in strips along the foothills and contours. This indicates that in the urban zone, which is mainly influenced by economic activities and increased demand for construction land, farmlands were converted on a large scale into patches and clusters. In the mountainous areas, where human activities are relatively low, farmland areas were reduced in the form of strips due to the impact of the reforestation project.

#### 3.2.3. Interaction Factor Analysis

The interaction detection reflects the difference in the effect of the joint action of factors on land use change relative to the action of single factors, and the results of the interaction detection are shown in Figure 11. The most interaction detection results of the driving factors of each type of land showed two-factor enhancement or non-linear enhancement, and there was no independent or weakened situation, which indicates that the explanatory power of the interaction between factors on land use change was enhanced to different degrees compared to that of a single factor and confirms that land use change is a complex process of factor interaction.

For farmlands, the explanatory power of the interactions between GDP and DEM, GDP and temperature were 0.51 and 0.46, respectively, indicating that the three factors, i.e., GDP, DEM, and temperature, jointly promoted the flow of farmlands during this period. For woodlands, the strongest explanatory power of the interaction between GDP and precipitation (0.34) indicates that GDP and precipitation were important factors in the transformation of woodlands. For grasslands, the interaction between temperature and precipitation had the greatest explanatory power (0.35), and the interaction between GDP and precipitation and GDP and temperature had explanatory powers of 0.32 and 0.30, respectively, reflecting the direct influence of precipitation and temperature on grasslands. For watershed areas, the interaction between temperature and slope and temperature and precipitation had explanatory powers of 0.031 and 0.032, respectively, reflecting the three most sensitive factors affecting watershed conversions. For construction land, the interaction between GDP and precipitation and temperature and slope was the strongest, and the interaction between DEM and GPD and precipitation was also strong, indicating that the main direction of urban expansion is into areas with flat terrain, small diurnal temperature difference, and abundant precipitation, in terms of unused land. The transformation characteristics could not be highlighted because of the small area, so therefore it was not repeated.

Overall, the GDP, precipitation, temperature, and slope were identified as the main factors affecting the land use transformation in the Zhengzhou metropolitan area.

### 3.3. Land Use Change Simulation

The CA–Markov model in IDRISI software was used to simulate and predict the land use types in the study area in 2015, in which the simulated baseline data were used in 1995 and 2005. The simulation results were compared with the actual 2015 land use type data to verify the accuracy of the modelled simulation. The comparison (Figure 12) shows that the simulation accuracy of all five land use types, except for unused land, was significant.

In addition, this study used the CROSS-TABULATION function in IDRISI software to further verify the accuracy of the model. By comparing the simulation data of the study area in 2015 with the remote sensing data, the model simulation accuracy table of land use types in 2015 was obtained (Table 5), and it is known from the Kappa coefficient classification evaluation criteria that 0.8 < 0.9165 < 1, which indicates that the simulation accuracy of the model was significant, and the simulation results were consistent with the real situation. Therefore, the model can be used for pre-simulation prediction of land use types.

Since the modelled simulation was verified to be adequately accurate, the land use map of the Zhengzhou metropolitan area in 2015 was used as the base year to integrate the suitability atlas of farmlands, woodlands, grasslands, water areas, construction land, and unused land, based on MCE. The CA–Markov model was used to simulate and predict the land use types in the study area.

According to the simulation results (Figure 13), the area of cultivated land in the study area decreased rapidly during the period of 2015–2035, whereas the construction land showed an increasing trend, and other land types did not change considerably. From the table of land use changes (Table 6), the areas of farmlands, woodlands, grasslands, water, construction land, and unused land in the study area in 2025 will be 22,416.53 km², 1644.15 km², 1462.40 km², 1047.90 km², 7845.704 km², and 6.783 km², respectively. It was also predicted that the areas of farmlands, woodlands, grasslands, water, construction land, and unused land in the study area in 2030 will be 21,482.14 km², 1392.80 km², 1093.30 km², 1362.04 km², 9089.42 km², and 3.78 km², respectively. The areas of farmlands, woodlands, grasslands, water, construction land, and unused land in the study area in 2035 will be 21,052.41 km², 1392.80 km², 1093.30 km², 1362.04 km², 9089.42 km², and 3.78 km², respectively.

The land use area transfer matrix produced by the simulation results (Table 7) shows that the area of farmlands will decrease considerably and that of construction land will increase markedly between 2015 and 2035, while the overall change of other types of land use will not be significant. Among these, farmlands will decrease from 23,679.37 km² in 2015 to 21,052.41 km² in 2035, with a relative change of 2626.96 km² and a total percentage decrease of 7.63%, which will be the land use type with the largest area decrease. It indicates that a large amount of farmland resources will be occupied in the process of urbanization in the study area during this period; however, from 2030, the observed decreasing trend of farmland resources will slow down markedly due to the farmland red line, which indicates that the implementation of the policy will effectively protect the farmland resources, which will still be the land type with the largest area. Construction land will increase from 6590.93 km² in 2015 to 9157.44 km² in 2030, with an overall increase of 7.45%, indicating that the urbanization of the Zhengzhou metropolitan area will steadily progress. Woodlands are predicted to decrease from 1691.96 km² in 2015 to 1381.37 km² in 2030, mainly due to the provision of land resources for the development of farmlands and construction land. Grasslands are projected to increase from 1490.03 km² in 2015 to 1517.60 km² in 2030, the main source of which may be land retired from cultivation and the increase in the level of urban greening. The area of water areas will increase from 964.22 km² in 2015 to 1307.20 km² in 2030, with a relative change of 342.99 km² and 1% increase in the total area share. The area of unused land is estimated to increase from 6.98 km² in 2015 to 7.46 km² in 2030, with a relative change of 0.48 km².

Figure 14 shows that the eastern part of the study area will rapidly expand into the construction land, cultivated land will slowly decrease, and woodlands and grasslands will gradually develop into cultivated land and construction land. Among these, the areas of farmlands, woodlands, and grasslands will be effectively protected due to the restrictions of the farmland and ecological red line. The expansion of construction land is predicted to be centered around five prefecture-level cities in the eastern and southeastern regions, while Zhengzhou city will expand the fastest to the east and south, a little slower to the west, and rarely to the north, mainly due to the physical restriction of the Yellow River channel, which makes it difficult for the city to develop across the river.

The remaining four cities are also developing rapidly. Predictions show an expansion pattern radiating outward from the core urban area, with the fastest urbanization in the township area around Xinxiang, characterized by a ribbon distribution along the northern bank of the Yellow River channel, and the southern bank of Zhengzhou and Kaifeng, presenting a state of separation from each other. The western and northern parts of the study area will be affected by the topography, such as the slope and height. The expansion rate of building land in some areas are projected to lag that of the eastern flat terrain areas, but woodlands and grasslands will be better protected. The Yellow River course crosses the entire study area from west to east in the middle, and the expansion of towns along the banks of the Yellow River will be markedly higher, indicating that the cross-river development strategy proposed by Zhengzhou will be gradually implemented following the formation of the metropolitan area concept. The planning, using Jinan’s cross-river development as a reference, will gradually improve.

Compared to the spatial disadvantage of urban expansion in Jinan, which is located south of the Mountain Tai and backed by the Yellow River, Zhengzhou is relatively open and flat on three sides, but with further urbanization Zhengzhou is bound to expand north of the Yellow River; the course of development across the river is an important issue that cannot be avoided in future land planning. At the same time, the construction land within the study area has the characteristics of agglomeration, regular distribution of construction land, blurred boundaries between towns, and gradually tending to be integrated, which is also consistent with the overall development strategy of the “ZhengBian integration” and “Zhengzhou metropolitan area integration”.

## 4. Discussions

The land use system is a complex system. This study constructed relevant factors affecting land use change from land adaptation variables and socio-economic variables; however, there still remains some difficulty in quantifying certain factors. Different choices of variables and indicators can lead to differences in simulation results. There-fore, the adequate selection of more scientific impact indicators is of great significance for future research in this field. In this study, between 1980 and 2015, there has been a significant shift in construction land use in Zhengzhou, with a large portion of farmland replaced by construction land; a similar situation was observed in Beijing from 1978 to 2013 [71], Shenzhen from 1988 to 2015 [72], and most cities in China [10,15,73]. This suggests that economic development takes precedence over agricultural development in urban expansion. Between 2000 and 2005, construction land in the Zhengzhou metropolitan area grew rapidly (Figure 2). In addition, between 2010 and 2015, the main urban area of Zhengzhou experienced a decline in construction land and increase in green space. Construction land in the surrounding cities of Zhengzhou, such as Zhongmu and Xingyang, maintained an increasing trend, suggesting that the expansion of urban space driven by policy planning during urbanization of Zhengzhou was completed at this stage, and the government began to focus on greening and environmental improvement of the city. Other cities will maintain the rate of expansion due to slower urbanization. The urban expansion pattern of the Zhengzhou metropolitan area shares characteristics with the early development of Chinese cities. From 2000 onwards, government departments began to work on planning for larger urban agglomerations; urban agglomerations near ports and rivers, such as those in the Yangtze River Delta [74,75,76] and Pearl River Delta [77], all showed a tendency to expand along rivers and near ports in the early years of urban expansion. The Zhengzhou metropolitan area is in the central region and has flat topography, which makes it cheaper to transform unused land and farmlands into urban areas, which can effectively reduce construction costs and facilitate urban expansion.

Land use in the Zhengzhou metropolitan area is influenced by a combination of factors, such as population growth, economic development, urbanization, ecological protection, land use policies, topography, slope, and climate change. The degree of influence of different factors on land use change varies greatly. In this study, it is argued that socioeconomic development is the dominant driver of land use conversion into construction land in the Zhengzhou metropolitan area. These results are consistent with previous studies [62,78]. The statistics in this study show that from 1980–2015, the population and economic growth in the Zhengzhou metropolitan area showed a significant positive correlation with the expansion of built-up land. The results of the PLUS analysis further indicate that the expansion of built-up land was significantly (*p* value is 0) associated with population and economic growth. Figure 10 shows that the areas with relatively rapid expansion of built-up land and economic growth were mainly located in Zhengzhou, Kaifeng, Xinxiang, and the strip between Zhengzhou and Xuchang. The areas with relatively slow land expansion and economic growth in the built-up area were mainly located in Jiaozuo and the western part of Zhengzhou.

Since 1980, the population of Zhengzhou has grown rapidly, and as the economy has grown and urbanization has increased, the rural population has gradually migrated to the city center. The population growth within the city has reduced the land area per capita, and the new population inevitably requires new carrying space, thus generating a strong demand for construction land and driving the rapid expansion of built-up areas [79]. Further analysis showed that the driving force of population growth on the expansion of construction land is gradually weakening from the core of Zhengzhou outwards. In contrast, economic growth has been the main driver of urban construction land expansion. High elevation has a large impact on the conversion of farmlands to woodlands in the southwestern region. This is closely related to the policy of returning farmlands with slopes greater than 25° to forests, implemented by the Henan provincial government since 2000 [10,80]. In addition, temperature and precipitation are important drivers of the conversion of grasslands to farmlands in the Zhengzhou metropolitan area. During these 35 years, the beach lands along the north and south sides of the Yellow River were converted to farmlands. The Yellow River is the main source of freshwater in the Zhengzhou metropolitan area, and the farmlands are extremely dependent on water resources; the degree of water resources exploitation directly affects the change in the trajectory of the expansive farmlands. In terms of elevation, the lower plains in Henan province are suitable for agricultural production, and the lower the elevation, the more pronounced the expansion of farmlands. The economic development of the areas along the Yellow River is generally higher than that of other regions, particularly in Zhengzhou and Xinxiang, which are important cities in the Central Plains Economic Zone. The demand for construction land continues to increase, causing the stripping of a large amount of farmland resources.

Meanwhile, rapid economic development brought about by accelerated fixed asset investment is a direct driver of land use change in China [81]. Construction land is an important component of fixed asset investment [82], and its source has increased the intensity of land use and, on one hand, expanded urban space and, on the other hand, transformed from larger areas of farmlands. In terms of industrial structure, from 1980 to 2015, the proportion of tertiary industries increased considerably, and the proportion of primary and secondary industries decreased markedly. The outward shift of the secondary industry and adjustment of the internal structure of the industry brought about the transformation of the land use structure and expansion of construction land, which further increased the conversion rate of farmlands.

Through simulation analysis, it was determined that the total farmland area in the Zhengzhou metropolitan area will decrease by approximately 2626 km² in 2035, with the main conversion being farmlands to construction land (and the conversion among other types of land uses will be in a relatively balanced state). The main regions where farmland changes are predicted are in the southern part of Xinxiang and township areas in the southeastern part of Zhengzhou. Since 2018, China has proposed and implemented a rural revitalization strategy, which sets long-term goals for the comprehensive revitalization of rural industries and promotes changes in the structure and function of rural land use. The rural revitalization strategy optimizes the spatial land use structure in rural areas, promotes the flow between lands, accelerates the economic development of rural areas, and improves urbanization. Zhou et al. [10] also reported similar findings. The intervention of land policies will lead to a significant increase in the level of construction land in the future Zhengzhou metropolitan area, but the reduction of farmlands also poses a potential problem for food security. These findings can assist governments to understand the future land use changes in the Zhengzhou metropolitan area according to the natural development scenario and thus support land use decisions.

This study also has limitations that should be explored in future research. Construction land use change is a very complex activity that involves consideration of economic, social, governmental planning, and ecological governance factors. Firstly, although meta-cellular automata and Markov chains are more applicable model simulation methods, the internal mechanism of land use change is extremely complex, which contains far more influencing factors, such as population density and flow, GDP, policies, soil types. Secondly, for the selection of model parameters, a generalized 30 × 30 meta-cell size and 5 × 5 filter were used without considering the influence of the meta-cell size and filter on the prediction results. Thirdly, the entire study area was considered as a whole unit, and the simulation errors caused by internal differences were ignored. Lastly, although the methodology and results of this study can serve as a reference for similar work, it does not fully capture the special factors that characterize the development process of other cities, particularly coastal areas and areas with higher economic development. Subsequent studies are required to further analyze other factors and variables that influence the development of urban construction and determine the mechanisms of construction land expansion. As urban agglomerations grow, government planning becomes more influential in changing construction land use, while other growth and influencing factors become more complex. Factors affecting the expansion of construction land, such as the policy system, industrial structure upgrading, and urban planning, cannot be precisely quantified. Future studies should analyze the internal influencing factors by combining statistical analysis of data with field research to achieve better spatial prediction effects and provide a more reliable basis for rational land use planning and ecological protection.

## 5. Conclusions

Due to the relatively stable natural environment in the Zhengzhou metropolitan area, natural drivers will only slowly influence land use changes in the area over a long period. Combined with the actual situation, when exploring the drivers of land use change in the study area, only a quantitative analysis of drivers was performed for those criteria that could be quantified in the socioeconomic drivers. The results of this study found that arable land and construction land in the Zhengzhou metropolitan area have changed substantially between 1980 and 2015. Among these, the area of arable land decreased each year because large portions were converted into forestlands and construction land due to urbanization and the policy of returning farmland to forests. The factors affecting the conversion of arable land mainly include the population and GDP. In the simulation results, this study determined that the relationship between urban construction and agriculture will be challenged by the excessive expansion of construction land and a sharp decline in farmlands. The forecast results show that the overall land use structure in 2035 will still be dominated by farmlands and construction land, while the areas of woodland, grassland, water land, and unused land will tend to be stable and almost unchanged. The area of farmland will be relatively reduced, but the overall area could still be maintained at a certain level due to the protection provided by the basic farmland policy. The growth trend of construction land was more evident. The simulation showed that land use changes will be more active in the future, and the urbanization level of the study area will be higher.

From the above analysis, the strongest influence on land use change was determined to be the increase in the urbanization rate, resident population, or non-agricultural population. Among these, the influence of non-agricultural populations on the evolution of construction land was more evident, indicating that the increase in the urbanization level will cause the encroachment of construction land on agricultural land. At the same time, the increase in the urban resident populations can squeeze the original construction land, thus further promoting the change of construction land production. This study can provide a reference for land use decisions in urban metropolitan areas.

## Figures and Tables

**Figure 1 ijerph-19-14089-f001:**
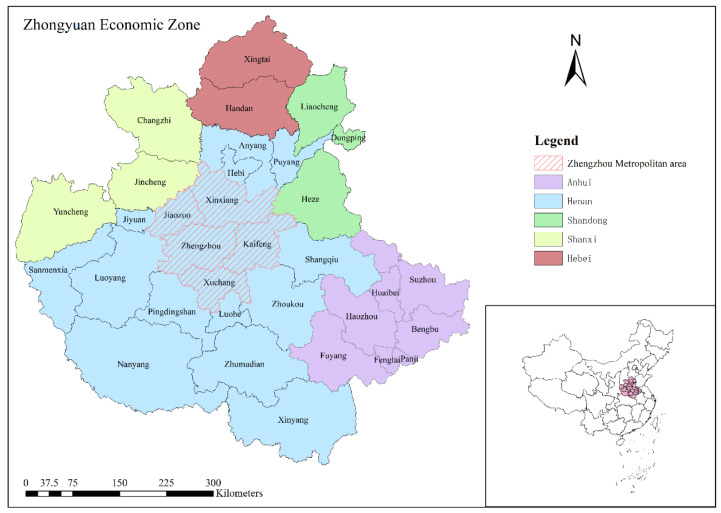
Study Area.

**Figure 2 ijerph-19-14089-f002:**
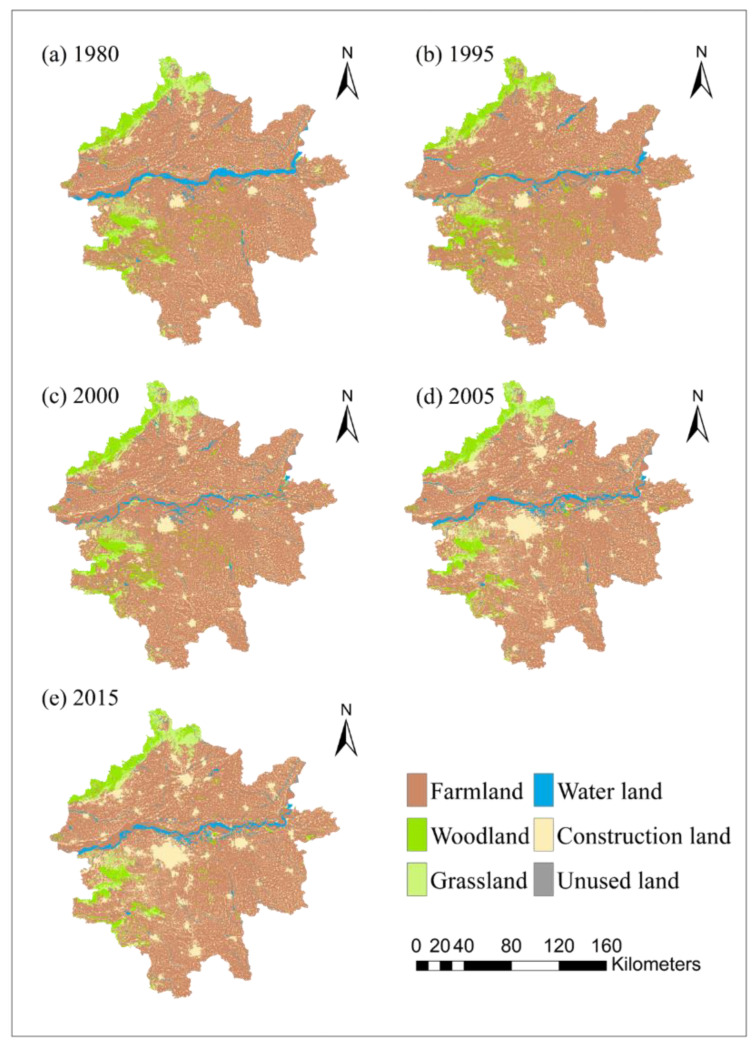
Distribution results of land use types in (**a**) 1980, (**b**) 1995, (**c**) 2000, (**d**) 2005, and (**e**) 2015.

**Figure 3 ijerph-19-14089-f003:**
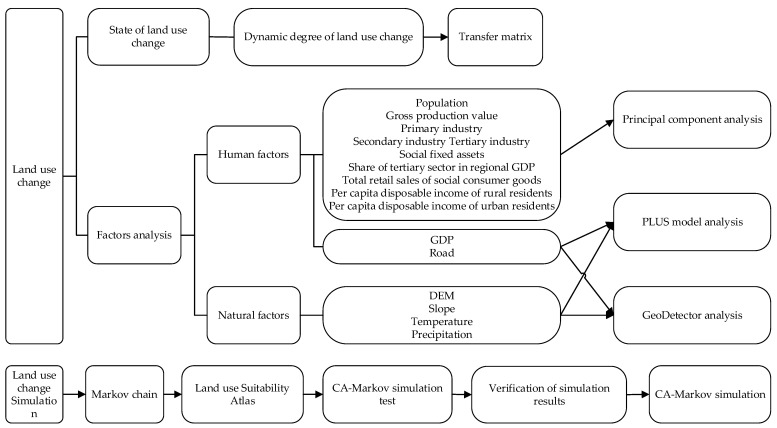
Analysis framework.

**Figure 4 ijerph-19-14089-f004:**
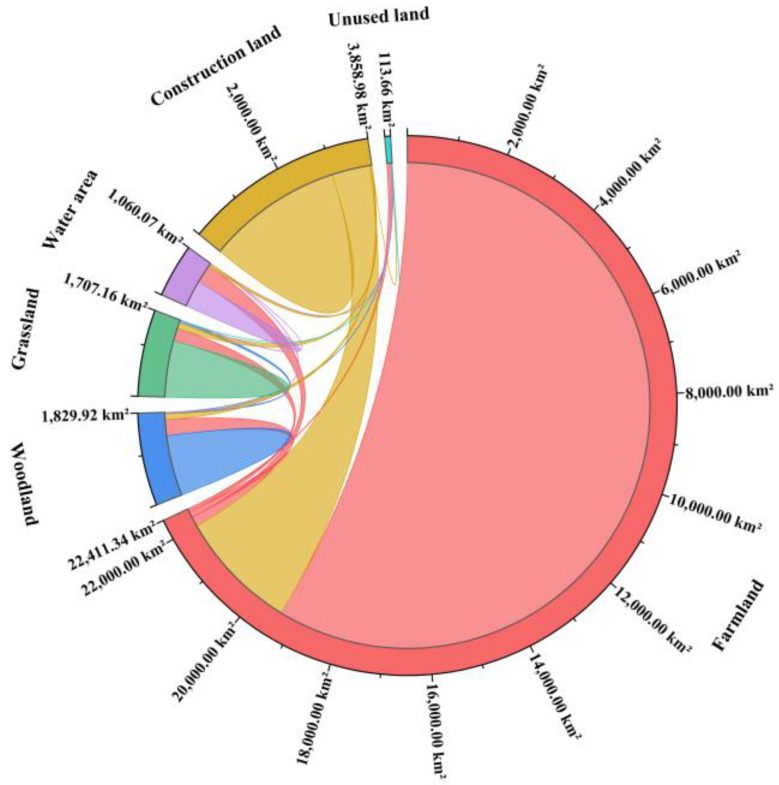
Land use change in Zhengzhou Metropolitan from 1980 to 2015.

**Figure 5 ijerph-19-14089-f005:**
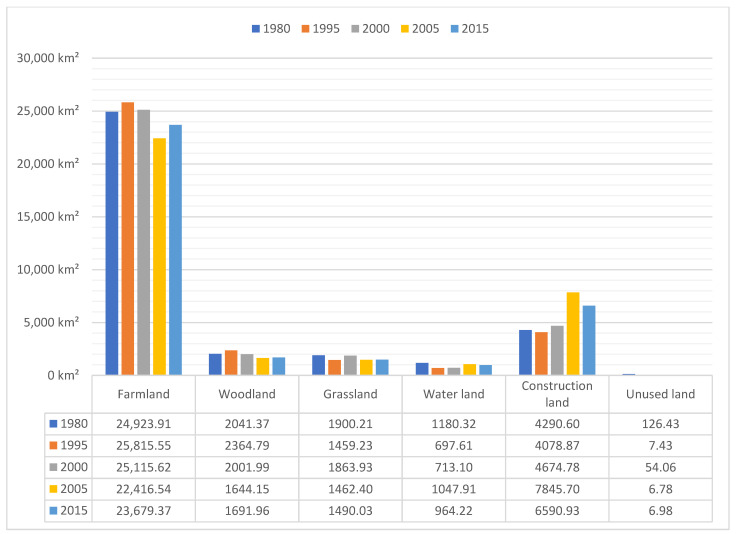
Land use change from 1980 to 2015.

**Figure 6 ijerph-19-14089-f006:**
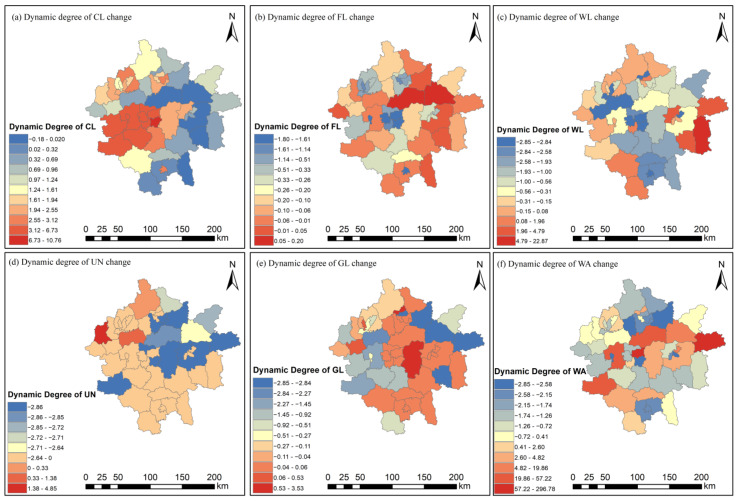
Spatial distribution of dynamic land use attitudes of (**a**) construction land (CL), (**b**) farmland (FL), (**c**) woodland (WL), (**d**) unused land (UN), (**e**) grassland (GL), and (**f**) water area (WA).

**Figure 7 ijerph-19-14089-f007:**
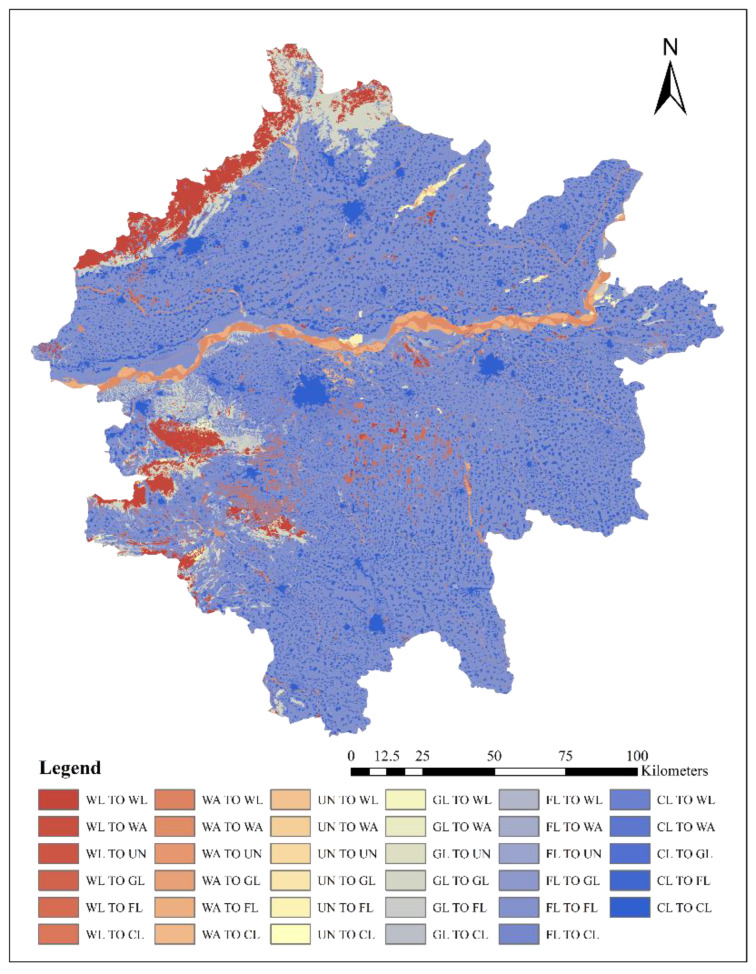
Spatial distribution of land use changes in Zhengzhou Metropolitan from 1980 to 2015.

**Figure 8 ijerph-19-14089-f008:**
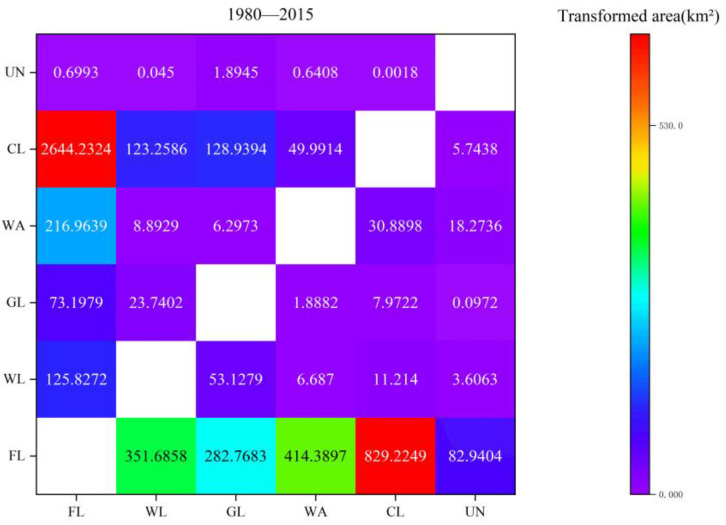
Land use change in different period of 1980–2015.

**Figure 9 ijerph-19-14089-f009:**
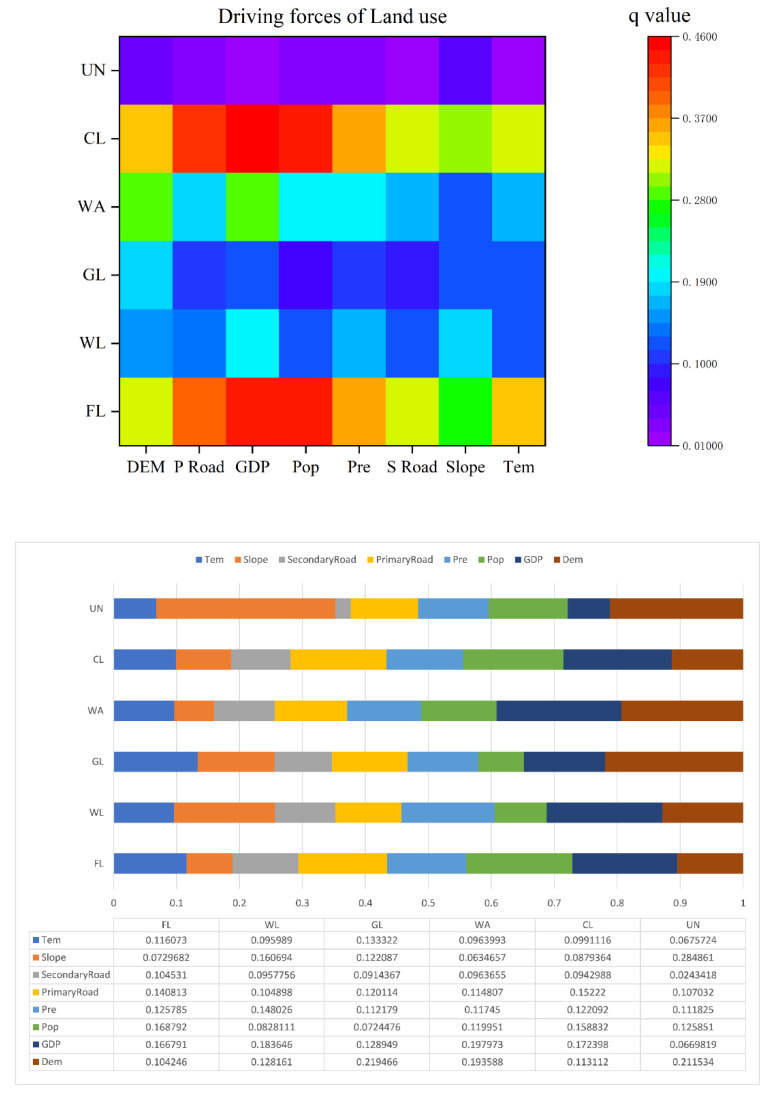
Driving forces of land use change from 1980–2015.

**Figure 10 ijerph-19-14089-f010:**
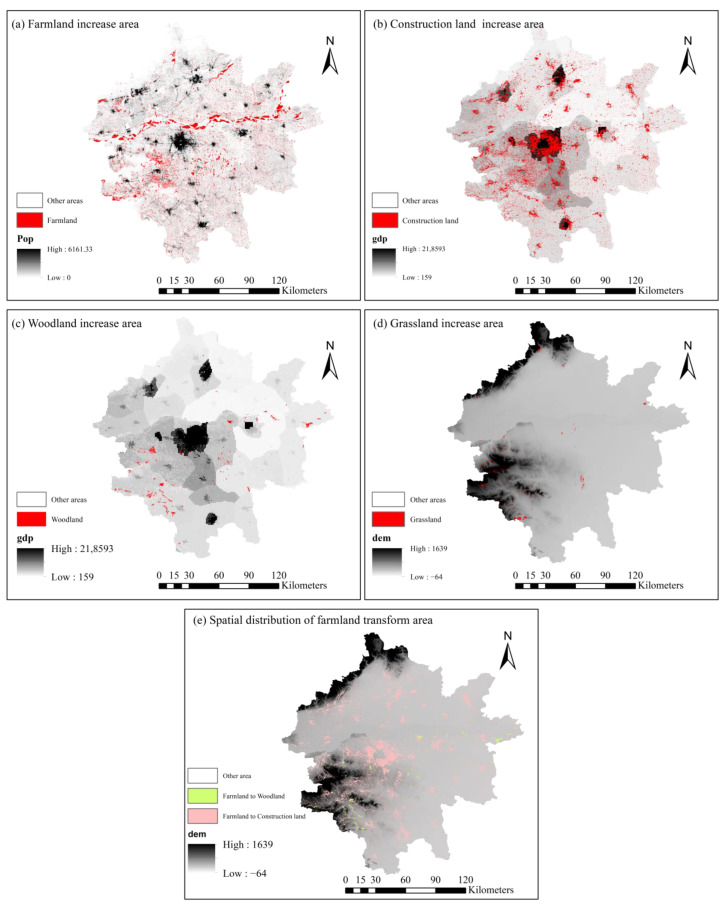
Land use-related factors of (**a**) farmland, (**b**) construction land, (**c**) woodland, (**d**) grassland, and (**e**) the spatial distribution of farmland transform area.

**Figure 11 ijerph-19-14089-f011:**
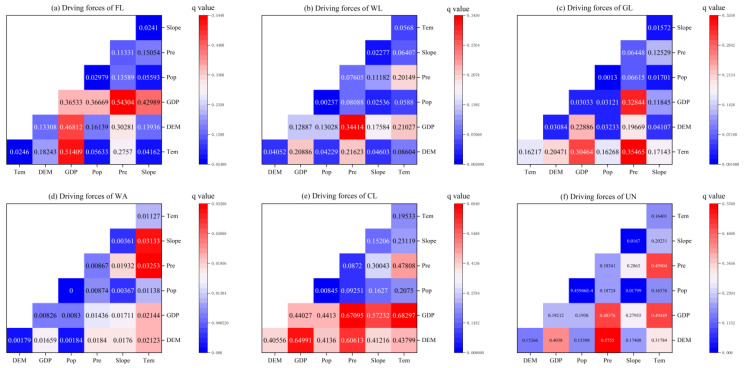
Interaction factors of (**a**) FL, (**b**) WL, (**c**) GL, (**d**) WA, (**e**) CL, and (**f**) UN.

**Figure 12 ijerph-19-14089-f012:**
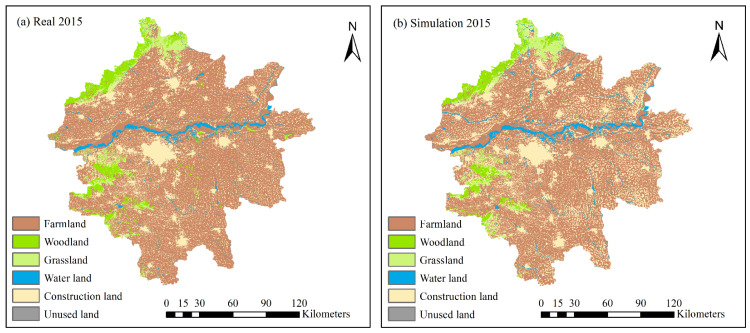
Comparison of land use from the (**a**) real and (**b**) simulated data in 2015.

**Figure 13 ijerph-19-14089-f013:**
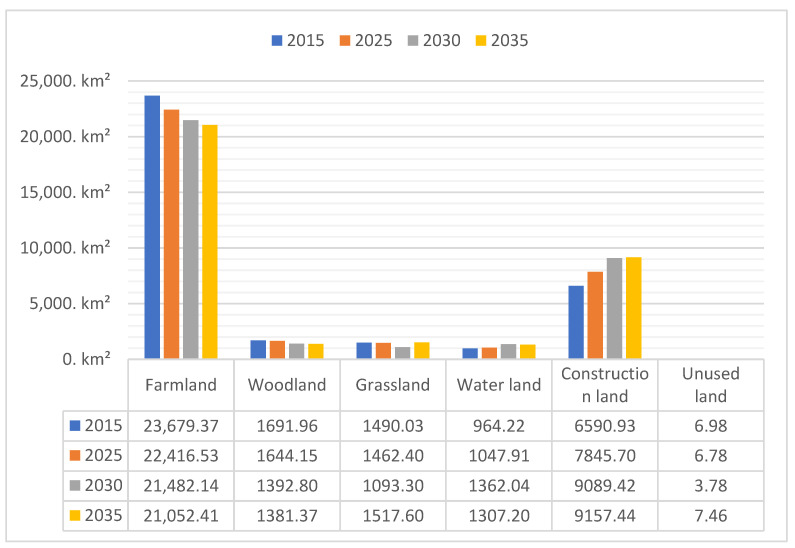
Land use forecast results for 2015, 2025, 2030, and 2035 (km²).

**Figure 14 ijerph-19-14089-f014:**
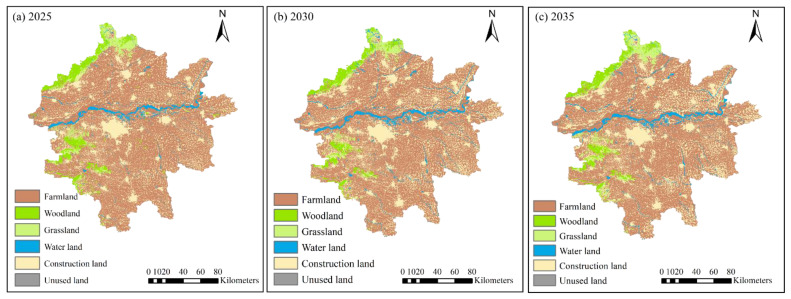
Land use simulation results for (**a**) 2025, (**b**) 2030, and (**c**) 2035.

**Table 1 ijerph-19-14089-t001:** Degree of spatial dynamics of land use types in different periods between 1980 and 2015 (unit: km²).

Land Type	1980–1995		1995–2000		2000–2005		2005–2015	
	Area Change	Dynamic Attitude	Area Change	Dynamic Attitude	Area Change	Dynamic Attitude	Area Change	Dynamic Attitude
Farmland	891.63	0.2303	−699.92	−0.5574	−2699.08	−2.4081	1262.83	0.5333
Woodland	323.42	0.9118	−362.79	−3.6243	−357.84	−4.3529	47.80	0.2825
Grassland	−440.98	−2.0147	404.70	4.3424	−401.53	−5.4914	27.63	0.1855
Water area	−482.71	−4.6130	15.48	0.4342	334.81	6.3901	−83.69	−0.8680
Construction land	−211.73	−0.3461	595.91	2.5495	3170.92	8.0832	−1254.77	−1.9038
Unused land	−119.00	−106.7447	46.63	17.2506	−47.28	−139.4073	0.20	0.2836

**Table 2 ijerph-19-14089-t002:** Transfer matrix of land use area from 1980 to 2015 (km²).

Time Period	Land Type	Farmland	Woodland	Grassland	Water	Construction Land	Unused Land	Transfer Out
1980–2015	Farmland	19,350.42	125.83	73.20	216.96	2644.23	0.70	3060.92
	Woodland	351.69	1322.30	23.74	8.89	123.26	0.05	507.62
	Grassland	282.77	53.13	1234.13	6.30	128.94	1.89	473.03
	Water area	414.39	6.69	1.89	586.48	49.99	0.64	473.60
	Construction land	829.22	11.21	7.97	30.89	2979.67	0.00	879.30
	Unused land	82.94	3.61	0.10	18.27	5.74	3.00	110.66
	Transfer in	1961.01	200.46	106.90	281.32	2952.17	3.28	
	Net transfer in	−1099.91	−307.16	−366.13	−192.28	2072.86	−107.38	

**Table 3 ijerph-19-14089-t003:** Results of the principal component analysis.

Ingredients	1	2
Total population (10,000 people)	0.8260	0.4794
Gross production value (billion yuan)	0.9977	0.0203
Primary industry (billion yuan)	−0.4669	0.8810
Secondary industry (billion yuan)	0.9932	−0.0994
Tertiary industry (billion yuan)	0.9934	0.0885
Social fixed assets (million yuan)	0.9996	−0.0102
Share of tertiary sector in regional GDP	0.7765	0.6107
Total retail sales of social consumer goods (million yuan)	0.9892	0.1195
Per capita disposable income of rural residents (10,000 yuan)	0.8866	−0.4615
Per capita disposable income of urban residents (10,000 yuan)	0.9775	−0.1709

**Table 4 ijerph-19-14089-t004:** Principal component load matrix of the driving factors.

Ingredients	Initial Eigenvalue			Extraction of the Sum of Squares of Loads		
	Total	Percentage of Variance	Cumulative %	Total	Percentage of Variance	Cumulative %
1	8.1912	81.9119	81.9119	8.19119	81.9119	81.91191
2	1.6537	16.5365	98.4484	1.65365	16.5365	98.44843

**Table 5 ijerph-19-14089-t005:** Land use simulation accuracy evaluation for 2015 (unit: km²).

Land Use Type	Actual Area	Simulated Area	Difference Area	Individual Accuracy
Farmland	23,679.37	21,090.58	2588.79	0.8215
Woodland	1691.96	1376.67	315.29	0.7631
Grassland	1490.03	1515.61	25.58	0.8705
Water area	964.22	1293.65	329.43	0.9979
Construction land	6590.93	9139.72	2548.79	0.9670
Unused land	6.98	7.26	0.27	0.5636
Overall accuracy (Kappa coefficient)	0.9165			

**Table 6 ijerph-19-14089-t006:** Land use change in 2015, 2025, 2030, and 2035.

Land Type	2015		2025		2030		2035	
	Area	Proportion	Area	Proportion	Area	Proportion	Area	Proportion
Farmland	23,679.37	68.79%	22,416.53	65.12%	21,482.14	62.41%	21,052.41	61.16%
Woodland	1691.96	4.92%	1644.15	4.78%	1392.80	4.05%	1381.37	4.01%
Grassland	1490.03	4.33%	1462.40	4.25%	1093.30	3.18%	1517.60	4.41%
Water area	964.22	2.80%	1047.91	3.04%	1362.04	3.96%	1307.20	3.80%
Construction land	6590.93	19.15%	7845.70	22.79%	9089.42	26.40%	9157.44	26.60%
Unused land	6.98	0.02%	6.78	0.02%	3.78	0.01%	7.46	0.02%

**Table 7 ijerph-19-14089-t007:** Transfer matrix of land use area from 2015 to 2035 (km²).

Time Period	Land Type	Farmland	Woodland	Grassland	Water Area	Construction Land	Unused Land	Transfer Out
2015–2035	Farmland	18,511.71	23.45	157.96	304.39	2313.90	0.02	2799.72
	Woodland	193.18	1176.26	24.23	7.37	121.71	0.02	346.51
	Grassland	46.28	37.68	1178.25	0.51	75.18	3.13	162.78
	Water area	4.36	0.04	0.22	857.78	5.40	0.00	10.02
	Construction land	205.59	0.86	2.70	2.50	5720.20	0.00	211.64
	Unused land	0.37	0.00	0.02	2.01	0.34	3.54	2.74
	Switch to	449.78	62.02	185.13	316.78	2516.52	3.17	
	Transfer in	−2349.94	−284.48	22.35	306.76	2304.88	0.43	

## Data Availability

(1) Land use data from 1980, 1995, 2000, 2005, and 2015 of the Henan province obtained from the Resource and Environmental Science and Data Center, Chinese Academy of Sciences (https://www.resdc.cn/, accessed on 1 July 2021); (2) ASTER GDEM V3 digital elevation at 30 × 30 m resolution, DEM, population, precipitation, temperature, and Henan provincial political subdivision data from the Geographic State Monitoring Cloud Platform (https://www.resdc.cn/, accessed on 5 July 2021); (3) administrative divisions, roads, rivers, and other auxiliary data from the Geographic State Monitoring Cloud Platform (https://www.resdc.cn/, accessed on 18 July 2021); (4) total population and GDP data of the study area from the statistical yearbooks from each time period (http://www.henan.gov.cn/, accessed on 19 July 2021).

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
