# Peer review of "Spatial-Temporal Changes and Simulation of Land Use in Metropolitan Areas: A Case of the Zhengzhou Metropolitan Area, China"

_ijerph, 2022, doi:10.3390/ijerph192114089_

Round 1
Reviewer 1 Report
Using remote sensing images and other auxiliary information, this study investigated the characteristics of land use change in Zhengzhou from 1980 to 2015 and analyzed its driving forces. Then, based on the CA-Markov model with multi-criteria evaluation suitability factor, the spatial and temporal change and evolution pattern of land use in 2015 were simulated, and the land use change in Zhengzhou from 2015 to 2030 was predicted. The paper is well-structured.However, there is a lack of methodological innovations and there are some concerns about the main concept, methodology, and data as follows.
Main comments:
1. The article points out that land use change has been a hot topic and scholars have conducted extensive and in-depth research on the on the spatial and temporal patterns, driving forces , and dynamic models of land use change. However, the studies cited by the authors are too old to support this view.
2. The land use data used in the author's article is from a long time ago, and the latest time is 2015, which is not conducive to accurately predicting future changes.
3. The author used the land use data of Henan Province in 1980, 1995, 2000, 2005 and 2015 in the study, but did not scientifically and reasonably explain the reasons for selecting the land use data of these different years.
4. In Chapter 3.1.2, the author uses the land use data of different time periods (1980-2005 and 2005-2015) to compare the size of land use change, which is not scientific.
5. The data in Figure 7 and Table 2 take 2000 and 2005 as nodes respectively, which are not the same. Why? Please give a reasonable explanation.
Minor comments:
1. In Chapter 3.2.2, the driving factors of the expansion of construction land and forest land cannot be analyzed from Figure 9, The analysis of the driving factors of the expansion of construction land and forest land should come from Figure 8.
2. Please indicate the specific year of the digital elevation, DEM, population, precipitation, temperature and other data used in the article.
Reviewer 2 Report
General comments
The manuscript intitled “Land use change and simulation in metropolitan areas: A case 2
of the Zhengzhou metropolitan area, China” described scientific findings showing a spatial and temporal variation in land use in the metropolitan area of Zhengzhou, China. The authors used 6 land classification and 5 different periods (from 1980 to 2015). The paper was well-planned and uses sufficient amount of data, but there is some points I would like to call attention.
Title: As the manuscript points to a spatial and temporal perspective, I think the authors could mention the temporal variation in the title, which is important for planning.
Abstract: The abstract reports clearly all aspects of the paper.
Introduction: The introduction reports all the important points and is well written. The authors begin the text from a broader perspective and tap into the home of the Zhengzhou metropolitan area.
Methods: The description of the Zhengzhou metropolitan area is complete. I believe that the authors could clarify the socioeconomic distribution in the metropolitan region. Aspects are important in land use change matters, however, there is a growing concern to consider economic and social aspects associated with the strategic planning of large and megacities. Items 2.2 through 2.4.3.4. are well outlined and explained.
Results: The results are consistent with the proposed objectives and methods. Figures 4 and 12 need a unit on the "Y" axis.
Figure 8 must also present unity on the X axis. Change the indication of the last image of figure 9 (change c for e).
Discussion: The discussion is substantiated and addresses the main results.
Conclusion: The conclusion is concise and brings elements that meet the objectives. The work is almost ready to be published.
Round 2
Reviewer 1 Report
This manuscript has revised the existing problems according to the opinions of the first review, but there are still some points to be improved:
1. I suggest that you show the highlights of the manuscript in the abstract.
2. Many of the methods were mentioned in this article. So I suggest that you add a technical roadmap to clearly illustrate your technical process.
3. In line 531, figure 11: “Constrction land” should be “Construction land”.
4. In line 599: “Compared to the spatial disadvantage of urban expansion in Jinan, which is located south of the Yellow Mountain and backed by the Yellow River” Yellow Mountain is in Anhui province(Mount Huang). Should it be Mount Tai? I suggest the author open the map and take a closer look at the location of Jinan.
5. In table1, the table should indicate the unit of measurement for “area change”, such as km2, Table 5 have a similar problem.
6. The time period in which the change occurred should be added in line 349(Figure 6). (For example:1980-2015)
Figure 7 Land use change in different period of (a)1980-2000, (b)2000-2015, (c)1980-2015. The image name does not correspond to the image content: I only see one picture.(1980-2015)There are no other two.(a) and (b)
